# Is It Possible to Diagnose Preoperatively a Tubal Ectopic Hydatidiform Molar Pregnancy? Description of a Case Report and Review of the Literature of the Last Ten Years

**DOI:** 10.3390/jcm11195783

**Published:** 2022-09-29

**Authors:** Marco D’Asta, Nicolò La Ferrera, Ferdinando Antonio Gulino, Carla Ettore, Giuseppe Ettore

**Affiliations:** Department of Obstetrics and Gynaecology, Azienda di Rilievo Nazionale e di Alta Specializzazione (ARNAS) Garibaldi Nesima, 95124 Catania, Italy

**Keywords:** molar pregnancy, ectopic pregnancy, salpingectomy, gestational trophoblastic diseases

## Abstract

Synopsis: Nowadays there are no clinical, laboratory, or ultrasound criteria to differentiate ectopic tubal pregnancy from tubal molar pregnancy, so a preoperative diagnosis is not possible. Objective: Tubal ectopic hydatidiform moles are a rare type of gestational trophoblastic disease. The aim of our work is to understand if it is possible to diagnose, preoperatively, a tubal ectopic molar pregnancy, starting from the evaluation of a complicated case report up to performing a review of the literature. Materials and Methods: A 27-year-old woman was referred to our department for right pelvic pain, vaginal bleeding, and positive beta-hCG (590 mUI/mL). At the ultrasound, the uterine cavity was empty and a unilocular cyst of 15 mm below the right ovary, suspicious of ectopic pregnancy, was described. Serial measurements of daily beta-hCG (2031 → 2573 → 3480 mUI/mL) and, after five days, a laparoscopic salpingectomy, were performed. The pathologist confirmed a diagnosis of “incomplete invasive vesicular mole with extrauterine implant”. A review of the literature was performed, following the PRISMA statement, and searching all the articles related to this topic in the last ten years from PUBMED. We obtained data from thirteen studies, describing fourteen cases. Discussion: Considering the data from the literature, the main clinical symptoms were pelvic pain (100%), vaginal bleeding (64%), vomiting (7%), and fever (7%). By ultrasound examination, left adnexal mass on ten women (72%), and right adnexal mass on four (28%), were described. An assessment of ectopic pregnancy was made in all cases, but no preoperative diagnosis of tubal molar pregnancy was made. Beta-hCG levels were the same as patients with ectopic tubal pregnancy. Conclusion: Nowadays there are no clinical, laboratory, or ultrasound criteria to differentiate ectopic tubal pregnancy from tubal molar pregnancy.

## 1. Introduction

The hydatidiform mole is a type of gestational trophoblastic disease (GTD).

Gestational trophoblastic diseases (GTDs) are a spectrum of tumors and tumor-like conditions characterized by abnormal proliferation of pregnancy-associated trophoblastic tissue with progressive malignant potential. They are classified as premalignant and malignant diseases. Partial hydatidiform mole (PHM) and complete hydatidiform (CHM) mole are premalignant disorders; invasive mole, choriocarcinoma, a placental site trophoblastic tumor, and an epithelioid trophoblastic tumor (ETT), instead, are included among malignant diseases [1].

The complete mole arises from the fertilization of an empty ovum by spermatozoa with a haploid structure [2]. The partial mole instead arises from dispermic fertilization of an ovum with a haploid structure. The product is a cell with a triploid genome [3].

They are characterized by trophoblastic hyperplasia and focal or diffuse villous edema, with or without embryo in PHM or CHM, respectively. The invasive mole, instead, is characterized by villous myometrial penetration [1].

Partial or complete hydatidiform moles affect roughly 1 in 500 to 1000 pregnancies in Western countries [4]. However, tubal ectopic hydatidiform moles are quite rare lesions.

An ectopic pregnancy happens when a fertilized egg is sited and grows outside the uterus. This pathological condition usually occurs in a fallopian tube, which carries eggs from the ovaries to the uterus, and it is called a tubal pregnancy. In the scientific literature, there are only a few cases of tubal ectopic hydatidiform moles, so it is a diagnostic and clinical dilemma to diagnose and manage these conditions.

The aims of our study were to describe a case of tubal ectopic molar pregnancy and secondly to perform a narrative review of the literature to assess the clinical, diagnostic, and therapeutic aspects of this condition. The description of the case report was performed following the CARE criteria (https://www.care-statement.org/checklist, accessed on 1 August 2022).

## 2. Materials and Methods

### 2.1. Case Report

A 27-year-old para 0 secondigravida woman was referred to our hospital on 1 February 2020 with right pelvic pain, vaginal bleeding, and a positive beta-HCG value (590 mUI/mL). Her family history was negative for gynecological cancer. She did not have any previous personal or family history of molar pregnancy. She had not received any previous hormonal therapy or IVF. She did not smoke and she had not used any intrauterine device (IUD) in her life. In her previous medical history, in 2018, a left salpingectomy for left tubal ectopic pregnancy with hemoperitoneum was performed. The gynecological examination was negative. At ultrasound examination, the uterus and ovaries were normal. Below the right ovary, a unilocular cyst of 15 mm in size, with anechoic content and peripheral vascular ring, was described (Figure 1). The uterine cavity was empty (Figure 2). An endopelvic free fluid of 26 mm in size was visualized. An ectopic pregnancy was suspected, so the patient was admitted to the hospital for monitoring. Considering that since the admission there was no more pelvic pain and no vaginal bleeding it was decided to wait and to monitor daily her clinical conditions with serial measurements of beta-Hcg values (03/02 → 2031 mUI/mL, 04/02 → 2573 mUI/mL; 05/02 → 3480 mUI/mL). On the fifth day, considering the increasing value of beta-Hcg associated with ultrasonographic evidence of gestational sac in the right tube, surgical intervention was performed. During the laparoscopy, the right fallopian tube was enlarged and hyperemic, as for ectopic pregnancy. The right salpingectomy was performed. The uterus, the other ovary, the peritoneum, the bladder, and the other abdominal and pelvic organs were macroscopically normal. At the histopathologic examination, “incomplete invasive vesicular mole with extrauterine implants” was described. The patient was monitored up to a negative value of beta-HCG, reached after one month. BHCG title was monitored every two weeks until three consecutive months’ negative levels. A whole-body CT scan was also performed and there was not any sign of extraperitoneal dissemination.

### 2.2. Review of the Literature

A review of the literature was performed following the PRISMA statement (Preferred Reporting Items for Systematic Reviews and Meta-Analysis) [5]. We searched all the articles related to our topic in the last ten years from the international electronic bibliographic database PUBMED (from 1 January 2010 to 1 February 2021). The articles were found using comprehensive search criteria and a combination of MeSH terms. We used the following words for selection: (“molar pregnancy and tube,” “tube and ectopic pregnancy and mola,” “hydatidiform mole and tube,” “tubal mole pregnancy”). We selected the articles published between January 2010 and February 2021 (Table 1). We included articles concerning single case reports of tubal molar pregnancy where both treatment and the maternal prognosis were discussed. The search was limited to studies reported in the English language. The references of the items chosen were also evaluated for related citations. Two independent researchers assessed the titles and abstracts retrieved to select the most relevant articles. If the title and abstract did not provide enough information, the full text was obtained. Letters to editors, editorials, review articles, duplicates, and meta-analyses were excluded.

We only included items that followed our eligibility criteria, represented by pregnant women diagnosed with ectopic tubal pregnancy where histological examination revealed a molar pregnancy. We excluded those studies evaluating data from women with non-tubal molar ectopic pregnancy and those with heterotopic pregnancy, normal intrauterine gestation, and finally postmenopausal patients (Figure 3).

## 3. Results

We have performed a narrative review of the case reports described in the literature in the last ten years on tubal mole pregnancy. We obtained data from thirteen studies describing fourteen cases [6,7,8,9,10,11,12,13,14,15,16,17,18].

The age at diagnosis was 30 years old. The main clinical symptoms were abdominal pelvic pain (100%), vaginal bleeding (64%), vomiting (7%), and fever (7%).

At ultrasound examination, the left adnexal mass on ten women (72%) and the right adnexal mass on 4 (28%) were described with an empty uterine cavity. The endopelvic and abdominal free fluid was detected in 100% of women. A live fetus in two women (14%) was found [9,16]. In the other cases, a heterogeneous complex adnexal mass with medium/low echogenicity was described [18].

An assessment of ectopic pregnancy was made in all cases, and no preoperative diagnosis of tubal molar pregnancy was made in any case. Explorative laparoscopy and salpingectomy on five women (36%), laparotomy salpingectomy on seven (50%), and laparoscopy salpingostomy on two women (14%) were performed. In one of these, 100 mg methotrexate was injected into the right mesosalpinx, as a routine at the time of salpingotomy for ectopic gestation in that institution [18]. At histological examination, an ectopic complete molar pregnancy was confirmed in seven women (50%), a partial molar pregnancy on six (43%), and an invasive tubal mole with tubal rupture in one woman (7%). The treatment consisted of strict follow-up with serial blood workup, mainly monitoring of beta-hCG levels in thirteen cases (93%), including the invasive mole. In one case of partial mole, as elevated hCG levels were detected, after staging, with chest X-ray images and pelvic/transvaginal ultrasound, that was normal, chemotherapy was started. It consisted of 1 mg/kg intramuscular methotrexate (MTX) being started (on days 1, 3, 5, and 7), with 15 mg folic acid rescue therapy (on days 2, 4, 6, and 8) [11]. According to the literature data, the mean gestational age at diagnosis was eight weeks after the last period [19].

It is still accepted today that the symptoms of patients with ectopic molar pregnancies, such as abdominal pelvic pain and vaginal bleeding, and beta-hCG levels, are the same as patients with ectopic tubal pregnancies [20,21]. Ectopic pregnancy is usually associated with lower than average beta-hCG levels of a normal intrauterine pregnancy; instead, a molar pregnancy is usually associated with greatly elevated levels. In the case of a tubal ectopic hydatidiform molar pregnancy, we do not have any range of beta-hCG values to differentiate an ectopic tubal pregnancy from an ectopic tubal hydatidiform molar one. In our narrative review, and in some previous studies [20,21], the beta-hCG levels are the same in these two groups, but further studies are needed to have a reference range.

## 4. Discussion

Ultrasonography of GTD is a heterogeneous, hypoechoic, solid mass with cystic vascular spaces [22]. There are no ultrasound criteria for distinguishing ectopic tubal pregnancy from tubal molar pregnancy. The histopathological examination remains the gold standard for diagnosis [22]. It is an isolated phenomenon to have a histopathological diagnosis of hydatidiform mole in an ectopic tubal pregnancy, but it could happen [23].

Unfortunately, as demonstrated by our clinical case and by the results of the literature search, the increase of βhCG was similar between tubal ectopic pregnancies and tubal molar pregnancies, therefore it could not be considered a valid tool for the diagnosis.

A hydatidiform mole occurs as there is a vitality of trophoblastic proliferation in the first trimester [24]. Pathologically, hydatidiform moles are characterized by marked circumferential proliferation of trophoblast with the presence of hydropic degeneration in all or some of the placental villi. It is true that in the first part of the physiological placentation, the polar trophoblastic proliferation is present too, but the hydrops is absent or mild [21].

However, once confirmed, it is necessary to monitor the patient to rule out any development of choriocarcinoma, detectable only by signs and symptoms of metastatic disease; however, the likelihood of this possibility is uncommon [25]. Explorative laparoscopy, with or without salpingectomy, is considered the gold standard of treatment in ectopic pregnancies [26]. When a salpingectomy is performed, a histological diagnosis should be obtained so the treatment could be planned.

The first-line treatment for persistent GTD is based on methotrexate administration [27].

In cases untreated surgically, serial beta-HCG monitoring could reveal a tubal hydatidiform mole. In this case, appropriate counseling is needed to avoid compromising future treatment and prognosis [28].

The surgical choice of a salpingostomy vs. a salpingectomy has to be carefully evaluated when there is a diagnosis of a tubal molar pregnancy; considering the potential gravity of the evolution of a tubal molar pregnancy, a salpingectomy could be theoretically preferred, but we do not have enough data to confirm it. Therefore, it could be useful to evaluate for future research the RCTs of patients affected by tubal ectopic molar pregnancy treated by salpingectomy or salpingostomy, to show which has the better outcome.

In our clinical case, we have also tried to use 3D echography to make a diagnosis of this condition but it was not useful for the evaluation of the tube. Furthermore, we hope that new searches could identify characteristic ultrasound aspects of tubal ectopic molar pregnancy, as described for uterine GTD [26], that could differentiate it from a normal ectopic tubal pregnancy.

## 5. Conclusions

In conclusion, a tubal ectopic molar pregnancy is a relatively rare condition, difficult to identify. It can often be unrecognized until a histological examination is performed. In the literature, there are few cases. Nowadays there are no clinical, laboratory, or ultrasound criteria to differentiate ectopic tubal pregnancy from tubal molar pregnancy. Our experience has also been useful for the management of a woman with suspected hemodynamically stable ectopic pregnancy.

## Figures and Tables

**Figure 1 jcm-11-05783-f001:**
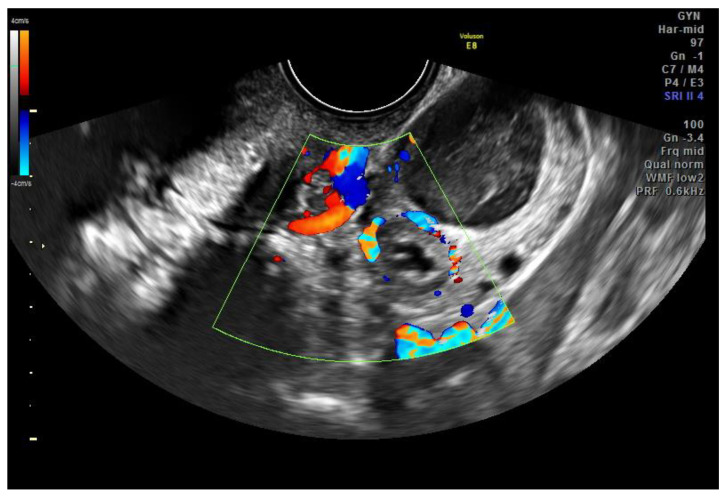
Ultrasound image of unilocular cyst of 15 mm in size.

**Figure 2 jcm-11-05783-f002:**
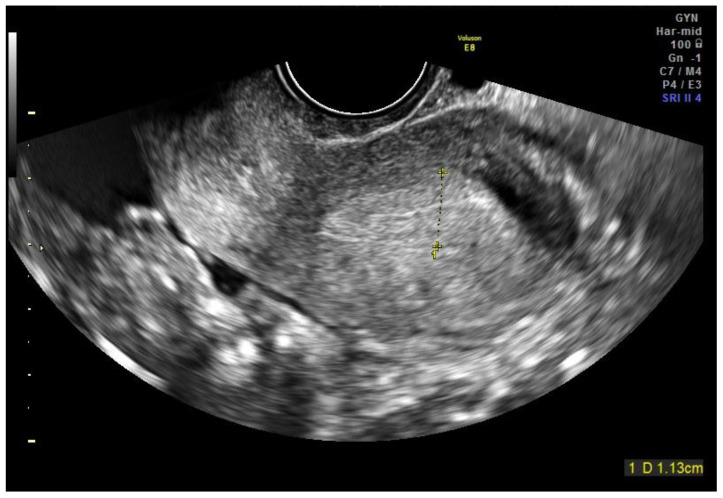
Ultrasound image of empty uterine cavity.

**Figure 3 jcm-11-05783-f003:**
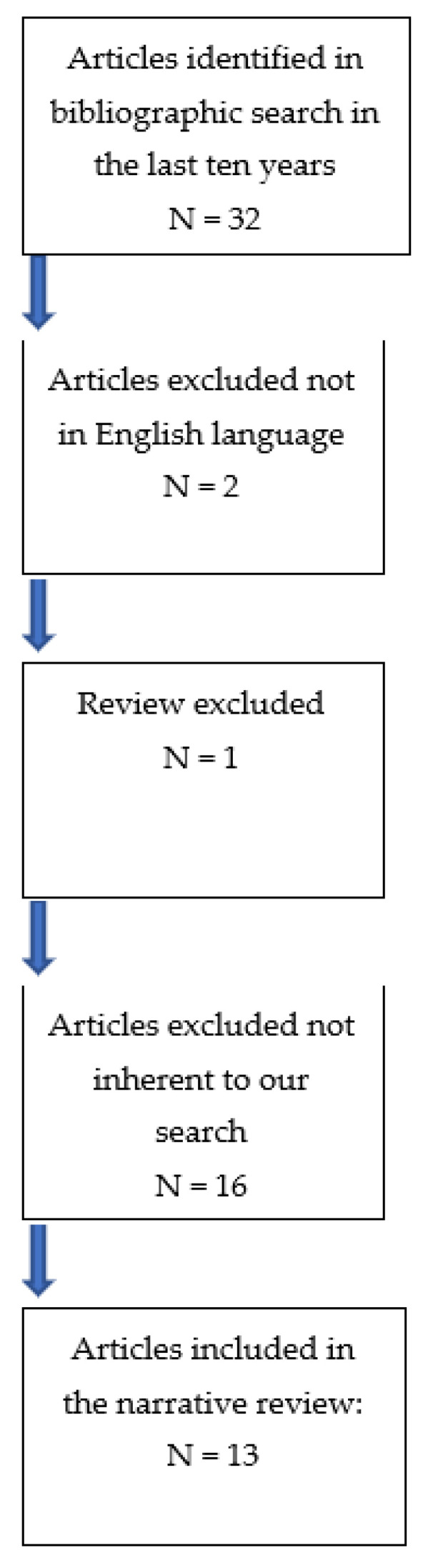
Articles included and excluded from the narrative review.

**Table 1 jcm-11-05783-t001:** Review of the literature of the last 10 years.

Author, Year	Study Design	Clinical Aspects	US	Treatment	Histological Examination	Follow-Up
Laura Allen et al., 2016 [6]	Case report	29 years BHCG of 32,000 IU/L. Diagnosis of a miscarriage one month before. Abdominal pain	Right-sided adnexal mass measuring 2.2 × 2.4 × 2 cm, and a fluid collection in the uterus. No evidence of a gestational sac	Salpingostomy and D & C	Partial hydatidiform molar (PHM) pregnancy	Serial monitoring of BHCG level; BHCG = 0 one month after surgery
A Siozos et al., 2010 [7]	Case report	β-HCG 3352 IU/L. Vaginal bleeding, abdominal pelvic pain	Left mass adjacent to the left ovary of 2.5 cm. Free fluid in the pouch of Douglas	Mini laparotomy with left partial salpingectomy	Complete molar pregnancy (CHM)	No symptoms 6 weeks later
Najoua Bousfiha et al., 2011 [8]	Case report	32 years BHCG 3454 IU/L. Last menstrual period 6 weeks before. Vaginal bleeding and lower abdominal pain	Irregular echogenic mass in the left adnexa (1.5 cm × 2 cm)	Left laparoscopic salpingectomy	Partial molar pregnancy (PHM)	Weekly quantitative Β-hCG titers until three successive Β-hCG levels were negative.
Borahe et al., 2010 [9]	Case report	30 years Last menstrual period 7 weeks before. Mild vaginal bleeding and pelvic pain	Left tubo-ovarian mass with a live fetus corresponding to 7 weeks and 6 days of gestation with free fluid in the pelvic cavity	Laparotomy left side salpingectomy	Complete molar pregnancy (CHM) and tubal rupture	Weekly follow-up by serum βhCG measurement
Chi-Wen Juan 2013 [10]	Case report	20 years BHCG 6984 mIU/mL. Last menstrual period 8 weeks and 4 days before. Abdominal pain	Empty endometrial cavity with cul-de-sac fluid and a left adnexal mass	Left laparoscopic salpingectomy	Tubal invasive mole and tubal rupture	Weekly quantitative β-hCG titers until 3 successive β-hCG levels were negative
Consuelo Lozoya López, et al., 2018 [11]	Case report	34 years BHCG 12,893 IU/L. Last menstrual period 8 weeks before. Abdominal pain, vomiting, and vaginal bleeding	Left paraovarian mass of 65 × 40 × 35 mm in size, filled with amorphous echoes	Laparotomy left side salpingectomy	Partial molar pregnancy (PHM) with a slight ruptured tube	Elevated hCG levels were detected. MTX was initiated
Haneen Al-Maghrabi et al., 2019 [12]	Case report	39 years BHCG 110.766 mIU/mL. Lower abdominal pain, abdominal distention, and low-grade fever for three days	Right adnexal heterogeneous complex mass (7 × 5 × 5 cm) and an adjacent right ovarian cyst (8 × 7 × 5 cm) with free fluid in the abdomen and pelvis	Laparotomy right salpingo-oophorectomy	Complete molar pregnancy (CHM) and tubal rupture	Follow-up by serum BHCG measurements
Tabassum Nakeer et al., 2014 [13]	Case report	32 years Abdominal pain and vaginal bleeding	Abdominal mass of 1.8 cm near to the left ovary and fluid in cul-de-sac	Laparotomy left side salpingectomy	Partial molar pregnancy (PHM)	
Devi Beena et al., 2016 [14]	Case report	32 years Last menstrual period one and a half months before. Abdominal pain and vaginal bleeding	Right adnexal mass 4 × 3 cm and endopelvic free fluid	Laparotomy right salpingectomy	Complete molar pregnancy (CHM) and tubal rupture	Weekly quantitative β-hCG titers until 3 successive β-hCG levels were negative
Fatemeh Davari Tanha et al., 2011 [15]	Case report	29 years BHCG 15,000 mIU/mL. Vaginal bleeding and pelvic pain	Left adnexal mass of 18 × 28 mm, free fluid in the cul-de-sac. No gestational sac in the uterus	Laparotomy left side salpingectomy	Partial molar pregnancy (PHM) with a slight ruptured tube	Serum beta-HCG titers
Chaouki Mbarki et al., 2015 [16]	Case report	32 years BHCG 40,400 mIU/mL. Last menstrual period 6 weeks before. Abdominal pain and mild vaginal bleeding	A left adnexal mass containing an embryo at 6 weeks of gestation with cardiac activity. No intrauterine gestational sac. Thin endometrium	Left laparoscopic salpingectomy	Partial molar pregnancy (PHM)	Weekly quantitative β-hCG titers until 3 successive β-hCG levels were negative
Chaouki Mbarki et al., 2015 [16]	Case report	37 years BHCG 290,600 mIU/mL. Last menstrual period 7 weeks before. Abdominal pelvic pain	5 cm left latero-uterine heterogeneous mass, a large pelvic effusion, and no intrauterine pregnancy	Left laparoscopic salpingectomy	Complete molar pregnancy (CHM) and tubal rupture	Weekly quantitative β-hCG titers until 3 successive β-hCG levels were negative
IA Yakasai et al., 2012 [17]	Case report	35 years Last menstrual period 12 weeks before. Abdominal pelvic pain	Well-encapsulated mass in the left adnexa, measuring 79.8 × 50 mm	Laparotomy left side salpingectomy	Complete molar pregnancy (CHM) and tubal rupture	Every 2 weeks quantitative β-hCG titers until 3 successive β-hCG levels were negative
Ting Zhao et al., 2019 [18]	Case report	27 years BHCG 6178 mIU/mL. Last menstrual period 4 weeks before. Abdominal pain	Right adnexal mass measuring 31 × 28 × 18 mm (medium/low echogenic), while no sac was detected in the uterine cavity	Right laparoscopic salpingotomy + 100 mg methotrexate injected into the right mesosalpinx	Complete molar pregnancy (CHM)	Weekly quantitative β-hCG titers until 3 successive β-hCG levels were negative

## Data Availability

Data is contained within the article.

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
