# Peer review of "Is It Possible to Diagnose Preoperatively a Tubal Ectopic Hydatidiform Molar Pregnancy? Description of a Case Report and Review of the Literature of the Last Ten Years"

_jcm, 2022, doi:10.3390/jcm11195783_

Round 1

Reviewer 1 Report

General Impression:

The authors have reported a case of tubal hydatidiform mole in a 27-year-old patient. Ectopic molar pregnancies are rare entities and -as highlighted in the manuscript- are hard to diagnose. The authors further reviewed the Medline database and summarized the findings of similar cases. Considering the rarity of this case scenario, the unknown etiology and risk factors of molar pregnancy, and the conducted literature review, I think this manuscript could be worth publication in the Journal of Clinical Medicine. However, considering the lack of information that concerns the medical history of the patient, the inappropriate manuscript structuring, and the considerable amount of grammatical mistakes, I believe this manuscript is not publishable in its current form and the authors should carry out extensive revisions to improve the quality of their work.

Comments:

1) The word "systematic" should be removed from the title since the authors did not follow a strict methodology of a systematic review on the one hand. On the other hand, the authors themselves stated that the review was narrative in nature within the text (lines 56, 111, and 120). 

2) In the abstract section (line 24), please replace the word (determination) with (monitoring) or (a serial measurement).

3) In the abstract section (line 26) and the case report section of the main text (line 79), please state the type of the hydatidiform mole that the patient had, i.e., whether it is complete or incomplete.

4) The introduction is mainly focused on the description of the conventional hydatidiform moles. I would suggest adding an explanation about ectopic pregnancy and relevant information related to ectopic molar pregnancies. The authors should not omit the fact that an ectopic molar pregnancy is also an ectopic pregnancy. This should shed light on the diagnostic dilemmas as well as the therapeutic complexities that could be imposed by such a condition.

5) In the introduction section (line 46), please write the word disease in the plural to make the sentence grammatically correct.

6) In the introduction section (line 58), I would suggest citing the website and including it in the reference list rather than writing it in the current form.

7) In the case report section, please mention the patient's gravidity and parity. In addition, it would be better to mention whether she had previous history of molar pregnancy or any of her relatives. I also think that declaring the presence or absence of ectopic pregnancy risk factors such as previous hormonal therapy, IVF, smoking, copper IUD, PID, etc...

8) In the case report section, please include any surgical or gross pathologic image of the ectopic molar pregnancy.

9) In the case report section (line 78), please write "microscopic examination", "histopathologic", or "pathologic" examination instead of "anatomopathological examination".

10) In the case report section (line 79), please clarify the meaning of "extrauterine implants". Does it refer to the ectopic implantation site of the molar pregnancy? Please ensure to state the type of the molar pregnancy, as previously recommended.

11) In the case report section (line 81), please correct the abbreviation "TC" to "CT". In addition, it would be better to write "A whole-body CT scan".

12) In the review of the literature section (lines 99 to 108), Please restructure this paragraph following a clearer context by convening the inclusion criteria and the exclusion criteria, and writing the inclusion criteria before the exclusion criteria. It is misleading in its current form.

13) In the review of the literature section (line 99), I believe excluding letters to the editors is a weakness point in this review because many authors tend to report such rare cases in the form of letters to the editor or brief communication. It would have been better to at least screen such articles instead of preliminary excluding them.

14) The results of the review should be included in the results section under a relevant subheading.

15) I found it very inspirational that some authors performed salpingostomy instead of salpingectomy for patients with tubal molar pregnancy. This gives tribute to the conventional management of intrauterine molar pregnancies. It is eye-catching that those patients had a similar recovery to those who had salpingectomy. I believe this sheds light on the dilemma of whether molar pregnancies should be treated with salpingectomy (like an ectopic pregnancy) or with salpingostomy with suction of conception products (like a conventional molar pregnancy). Additionally, salpingostomy is of utmost importance in patients with single tube desiring future fertility. Please discuss those points thoroughly in the discussion section. I would also discuss the potential role of proper irrigation and suction of the pelvic cavity to limit postoperative recurrence.

16) Table 2 should be re-designed. Please start with the overall articles that you had right after entering your search strategy. Please add in a different cell the articles you identified from screening the reference list of relevant articles. Then please indicate the number of articles that were eligible to be included based on title/abstract screening. After all, state the number of excluded articles with the reason for exclusion and the number of the article that were finally included.

17) In the discussion section, please implicate a proper citation style in this part of the text and avoid using abbreviations like (it isn't).

18) The written language should be revised as it contains a lot of typos and grammatical mistakes, especially in table 1.

Author Response

Dear Reviewer,

thank you for your efforts, it was a pleasure and an honour to receive your comments to improve our manuscript. We proceeded in modifying all the critical points following your suggestions and your comments. 

General Impression:

The authors have reported a case of tubal hydatidiform mole in a 27-year-old patient. Ectopic molar pregnancies are rare entities and -as highlighted in the manuscript- are hard to diagnose. The authors further reviewed the Medline database and summarized the findings of similar cases. Considering the rarity of this case scenario, the unknown etiology and risk factors of molar pregnancy, and the conducted literature review, I think this manuscript could be worth publication in the Journal of Clinical Medicine. However, considering the lack of information that concerns the medical history of the patient, the inappropriate manuscript structuring, and the considerable amount of grammatical mistakes, I believe this manuscript is not publishable in its current form and the authors should carry out extensive revisions to improve the quality of their work.

Thank you, it is an honour for us to receive your comments

Comments:

1) The word "systematic" should be removed from the title since the authors did not follow a strict methodology of a systematic review on the one hand. On the other hand, the authors themselves stated that the review was narrative in nature within the text (lines 56, 111, and 120). 

The word was removed

2) In the abstract section (line 24), please replace the word (determination) with (monitoring) or (a serial measurement).

It was replaced with serial measurement

3) In the abstract section (line 26) and the case report section of the main text (line 79), please state the type of the hydatidiform mole that the patient had, i.e., whether it is complete or incomplete.

It is an incomplete hydatidiform mole, it was inserted in the abstract section and in the main text

4) The introduction is mainly focused on the description of the conventional hydatidiform moles. I would suggest adding an explanation about ectopic pregnancy and relevant information related to ectopic molar pregnancies. The authors should not omit the fact that an ectopic molar pregnancy is also an ectopic pregnancy. This should shed light on the diagnostic dilemmas as well as the therapeutic complexities that could be imposed by such a condition.

A paragraph about ectopic pregnancy was added in the introduction section

5) In the introduction section (line 46), please write the word disease in the plural to make the sentence grammatically correct.

It was corrected (diseases)

6) In the introduction section (line 58), I would suggest citing the website and including it in the reference list rather than writing it in the current form.

Thank you for your suggestion, but we preferred to keep it in its current form considering that it is not an article but only a checklist to avoid being confusing for the readers

7) In the case report section, please mention the patient's gravidity and parity. In addition, it would be better to mention whether she had previous history of molar pregnancy or any of her relatives. I also think that declaring the presence or absence of ectopic pregnancy risk factors such as previous hormonal therapy, IVF, smoking, copper IUD, PID, etc...

A paragraph was added in the case report section including all these data

8) In the case report section, please include any surgical or gross pathologic image of the ectopic molar pregnancy.

Unfortunately, we have not retrieved in our archives a surgical image of this ectopic molar pregnancy, so we could not include it in the manuscript

9) In the case report section (line 78), please write "microscopic examination", "histopathologic", or "pathologic" examination instead of "anatomopathological examination".

It was written histopathologic examination

10) In the case report section (line 79), please clarify the meaning of "extrauterine implants". Does it refer to the ectopic implantation site of the molar pregnancy? Please ensure to state the type of the molar pregnancy, as previously recommended.

Extrauterine implants mean that there are not endometrial cells. It was added incomplete molar pregnancy

11) In the case report section (line 81), please correct the abbreviation "TC" to "CT". In addition, it would be better to write "A whole-body CT scan".

"A whole-body CT scan" was written

12) In the review of the literature section (lines 99 to 108), Please restructure this paragraph following a clearer context by convening the inclusion criteria and the exclusion criteria, and writing the inclusion criteria before the exclusion criteria. It is misleading in its current form.

In this section, we have restructured this paragraph to be clearer

13) In the review of the literature section (line 99), I believe excluding letters to the editors is a weakness point in this review because many authors tend to report such rare cases in the form of letters to the editor or brief communication. It would have been better to at least screen such articles instead of preliminary excluding them.

We have performed another check and re-evaluated if some letters to the editors or brief communication could been previously not considered for the review, but it did not happen

14) The results of the review should be included in the results section under a relevant subheading.

It was added a section for the Results of the manuscript

15) I found it very inspirational that some authors performed salpingostomy instead of salpingectomy for patients with tubal molar pregnancy. This gives tribute to the conventional management of intrauterine molar pregnancies. It is eye-catching that those patients had a similar recovery to those who had salpingectomy. I believe this sheds light on the dilemma of whether molar pregnancies should be treated with salpingectomy (like an ectopic pregnancy) or with salpingostomy with suction of conception products (like a conventional molar pregnancy). Additionally, salpingostomy is of utmost importance in patients with single tube desiring future fertility. Please discuss those points thoroughly in the discussion section. I would also discuss the potential role of proper irrigation and suction of the pelvic cavity to limit postoperative recurrence.

Thank you for your interesting suggestion: we have discussed those points in the discussion section

16) Table 2 should be re-designed. Please start with the overall articles that you had right after entering your search strategy. Please add in a different cell the articles you identified from screening the reference list of relevant articles. Then please indicate the number of articles that were eligible to be included based on title/abstract screening. After all, state the number of excluded articles with the reason for exclusion and the number of the article that were finally included.

Table 2 was re-designed following your suggestions

17) In the discussion section, please implicate a proper citation style in this part of the text and avoid using abbreviations like (it isn't).

It was corrected this section considering a proper citation style and avoiding using abbreviations

18) The written language should be revised as it contains a lot of typos and grammatical mistakes, especially in table 1.

An extensive revision of the article was made by an English native speaker, particularly in table 1

Best regards,

Dr. F. Gulino

Reviewer 2 Report

This case report and review of the literature focuses on the interesting and rare phenomenon of ectopic molar pregnancy. In its present form the article requires major revisions, as this reviewer feels that the data presented must be expanded upon in order for it to merit publication. In addition to the specific points made below this reviewer suggests separating the discussion section into separate results and discussion sections. Best of luck.

-          Line 64 – was histological analysis of the previous ectopic pregnancy available and if so what was the result of the analysis?

-          Line 70 – surveillance was the first line of management of the case. Were other options discussed with the patient? Were additional sonographic tests performed during surveillance and if so, what was seen? When the BHCG increased during surveillance was methotrexate treatment discussed and why was it rejected as a treatment option?

-          Methods: I suggest editing the diagram describing the selection process of the articles included in the study. In its current form the diagram supplied is confusing (it should start with the total number of studies found) – please refer to additional systematic reviews for guidance.

-          Table 1: This presentation of the available data is disorganised and very confusing. I suggest that this table is edited to include the following columns: Author (year), age of patient, gestationalo age according to last menstrual period, BhCG, presenting symptoms, procedure, findings during procedure, pathological analysis, follow-up. Additionally, use the same terms consistently as in the current state several terms are used interchangeably. Finally, all abbreviations must be explained under the table.

-          Discussion: when quoting means the standard deviation should be provided (age of patient). Weeks of gestation is not a normally distributed parameter therefore mean should not be used, median should be used with the interquartile range provided.

-          Line 145 – the average age of gestation was already quoted at the beginning of the discussion and therefore it is unnecessary to repeat it here.

-          Line 149 – The authors state “a higher tendency for rupture at the time of clinical onset”. How did you reach this conclusion? Could the authors elaborate on the evidence for this from previous study and their own study and the possible biological mechanisms for this finding?

-          Line152-153 – this sentence is unclear – was a preoperative diagnosis made or not?

-          The paragraph staring on line 15 “It isn’t an isolated phenomenon” – what isn’t? are the authors suggesting that we over-diagnose molar ectopic pregnancies? My personal professional experience does not support this statement and I would appreciate this point to be elaborated upon if it is indeed the point that authors meant to convey.

-          Line 157 – the authors describe the pathogenesis of molar pregnancy. Are there any biological factors that are different in ectopic molar pregnancies? This paragraph needs rephrasing as it is unclear what the authors are trying to convey.

-          In the conclusion the authors quote a trial not previously mentioned. As the subject of the paper is whether diagnosis of ectopic molar pregnancy is possible preoperatively I would suggest elaborating upon this paper and any other evidence in the discussion section.

-          Can BhCG levels be used to predict ectopic molar pregnancies? I request the authors to discuss this.

-          Other hallmarks of molar pregnancy include hyperemesis gravidarum, theca lutein cysts, hyperthyroidism and pre-eclampsia – is any data available regarding these conditions in ectopic molar pregnancies?

-          Treatment of ectopic pregnancies often culminates in surveillance until tubal abortion or methotrexate treatment with less than half of patients undergoing surgical treatment and histologic analysis of the specimen removed. How do the authors propose this affects the incidence ectopic molar pregnancies? How do they suggest from available data that molar pregnancy may be ruled out?  

-          Proof reading necessary as there are several grammatical and spelling mistakes, additionally some incomplete sentences (for example in Table 1 in the first row under the column US “a fluid collection in the uterus measuring.” With no measurement following).

Author Response

This case report and review of the literature focuses on the interesting and rare phenomenon of ectopic molar pregnancy. In its present form the article requires major revisions, as this reviewer feels that the data presented must be expanded upon in order for it to merit publication. In addition to the specific points made below this reviewer suggests separating the discussion section into separate results and discussion sections. Best of luck.

Dear Reviewer,

thank you for your efforts, it was a pleasure and an honour to receive your comments to improve our manuscript. We proceeded in modifying all the critical points following your suggestions and your comments. We have separated the discussion section in results and discussion

-          Line 64 – was histological analysis of the previous ectopic pregnancy available and if so what was the result of the analysis?

      The previous ectopic pregnancy was available but there were no signs of malignancy, therefore it was not included in the text

-          Line 70 – surveillance was the first line of management of the case. Were other options discussed with the patient? Were additional sonographic tests performed during surveillance and if so, what was seen? When the BHCG increased during surveillance was methotrexate treatment discussed and why was it rejected as a treatment option?

      The other options (expectant management and methotrexate) were discussed but the patient refused these options and preferred a surgical operation. The sonographic tests showed evidence of an ectopic pregnancy, but there was no evidence or signs which could get the suspicion of a tubal molar pregnancy

-          Methods: I suggest editing the diagram describing the selection process of the articles included in the study. In its current form the diagram supplied is confusing (it should start with the total number of studies found) – please refer to additional systematic reviews for guidance.

Table 2 was re-designed following your suggestions

-          Table 1: This presentation of the available data is disorganised and very confusing. I suggest that this table is edited to include the following columns: Author (year), age of patient, gestationalo age according to last menstrual period, BhCG, presenting symptoms, procedure, findings during procedure, pathological analysis, follow-up. Additionally, use the same terms consistently as in the current state several terms are used interchangeably. Finally, all abbreviations must be explained under the table.

      Table 1 was totally modulated and better organized to improve the quality of reading and comprehension for the readers. Unfortunately all these data (age of patient, gestational age according to last menstrual period, BhCG, presenting symptoms, procedure, findings during procedure, pathological analysis, follow-up) are not present in the considered works, therefore we were forced to include only some data

-          Discussion: when quoting means the standard deviation should be provided (age of patient). Weeks of gestation is not a normally distributed parameter therefore mean should not be used, median should be used with the interquartile range provided.

We have removed the terms mean

-          Line 145 – the average age of gestation was already quoted at the beginning of the discussion and therefore it is unnecessary to repeat it here.

      It was removed the first sentence at the beginning of the discussion to avoid repeating it

-          Line 149 – The authors state “a higher tendency for rupture at the time of clinical onset”. How did you reach this conclusion? Could the authors elaborate on the evidence for this from previous study and their own study and the possible biological mechanisms for this finding?

      Considering that we have not enough data to support and confirm this conclusion, we have removed this sentence

-          Line152-153 – this sentence is unclear – was a preoperative diagnosis made or not?

      We have modulated this sentence. A preoperative diagnosis was not made

-          The paragraph staring on line 15 “It isn’t an isolated phenomenon” – what isn’t? are the authors suggesting that we over-diagnose molar ectopic pregnancies? My personal professional experience does not support this statement and I would appreciate this point to be elaborated upon if it is indeed the point that authors meant to convey.

      Your suggestion is correct, it is not a common phenomenon. We have remodulated this sentence writing “It is an isolated phenomenon; however, it could happen to attend an over-diagnosis of hydatidiform mole in tubal pregnancy

-          Line 157 – the authors describe the pathogenesis of molar pregnancy. Are there any biological factors that are different in ectopic molar pregnancies? This paragraph needs rephrasing as it is unclear what the authors are trying to convey.

      There are not any different biological factors which differentiate molar pregnancies in comparison to tubal molar pregnancies, therefore we change this paragraph describing only the pathogenesis of molar pregnancy

-          In the conclusion the authors quote a trial not previously mentioned. As the subject of the paper is whether diagnosis of ectopic molar pregnancy is possible preoperatively I would suggest elaborating upon this paper and any other evidence in the discussion section.

      We have included this trial in the discussion section and we have removed it from the conclusion of the work.

-          Can BhCG levels be used to predict ectopic molar pregnancies? I request the authors to discuss this.

      We have added this aspect in the discussion section

-          Other hallmarks of molar pregnancy include hyperemesis gravidarum, theca lutein cysts, hyperthyroidism and pre-eclampsia – is any data available regarding these conditions in ectopic molar pregnancies?

      No, unfortunately, there are no available data about these conditions in ectopic molar pregnancies

-          Treatment of ectopic pregnancies often culminates in surveillance until tubal abortion or methotrexate treatment with less than half of patients undergoing surgical treatment and histologic analysis of the specimen removed. How do the authors propose this affects the incidence ectopic molar pregnancies? How do they suggest from available data that molar pregnancy may be ruled out?  

      Considering the available data in the literature the best follow-up for these conditions are represented by weekly quantitative β-hCG titers until 3 successive β-hCG levels are negative.

-          Proof reading necessary as there are several grammatical and spelling mistakes, additionally some incomplete sentences (for example in Table 1 in the first row under the column US “a fluid collection in the uterus measuring.” With no measurement following).

An extensive revision of the article was made by an English native speaker, particularly in table 1

Best regards,

Dr. F. A. Gulino

Round 2

Reviewer 1 Report

Dear Authors,

I have carefully reviewed the revised manuscript and I admit that it was significantly improved and all my concerns were addressed. However, I found one sentence unclear enough and could not understand the rationale behind it.

It is "tubal complete hydatidiform mole data from women with metastasis of complete tubal mole" in lines 110 and 111. Please explain its meaning and clarify why you excluded metastatic cases of tubal hydatidiform mole.

Otherwise, I have no further comments.

With kind regards,

Author Response

Dear Reviewer,

thank you again for your useful comment. We have removed this sentence because it was unclear; initially, we thought to exclude women with metastasis of complete tubal mole, but then we also analysed these patients in our literature review. Now I hope that it is clearer for the readers.

Kind regards

Reviewer 2 Report

I would thank the authors for their work on my previous comments and after reviewing the changes made would request the following minor revisions.

-          Line 50: “quiet” should be “quite”

-          Line 157: the authors state that it is accepted that beta-hCG levels are the same in ectopic molar pregnancy and ectopic non-molar pregnancy. I find this quite hard to accept without extensive discussion as classically ectopic pregnancy is associated with lower than average beta-hCG levels and molar pregnancy is associated with greatly elevated levels. I request the authors to add a discussion of this point, with the addition of evidence and biologic plausibility where possible, as in my opinion this is one of the key points when discussion diagnosis of both molar and ectopic pregnancies.

-          Line 164: This reviewer appreciates the attempt to clarify this sentence, but it still does not read clearly – I cannot understand if the authors mean to say that despite being an isolated phenomenon there is over-diagnosis? Or do the authors mean that there is underdiagnosis as the sonographic characteristics of molar ectopic pregnancy are indistinguishable from those of non-molar ectopic pregnancy?

Author Response

Dear Editor,

thank you for your efforts, we tried to improve our work with your suggestions-

  •          Line 50: “quiet” should be “quite”

The word was modified

  •  Line 157: the authors state that it is accepted that beta-hCG levels are the same in ectopic molar pregnancy and ectopic non-molar pregnancy. I find this quite hard to accept without extensive discussion as classically ectopic pregnancy is associated with lower than average beta-hCG levels and molar pregnancy is associated with greatly elevated levels. I request the authors to add a discussion of this point, with the addition of evidence and biologic plausibility where possible, as in my opinion this is one of the key points when discussion diagnosis of both molar and ectopic pregnancies.

Thank you for your comment, we have added a discussion of this point, which could be useful for future studies in this field. We have written: "Ectopic pregnancy is usually associated with lower than average beta-hCG levels of a normal intrauterine pregnancy; instead, a molar pregnancy is usually associated with greatly elevated levels. In the case of a tubal ectopic hydatidiform molar pregnancy, we have not any range of beta-hCG values to differentiate an ectopic tubal pregnancy from an ectopic tubal hydatiform molar one. In our narrative review and in some previous studies20-21 the beta-hCG levels are the same in these two groups, but further studies are needed to have a reference range."

  • Line 164: This reviewer appreciates the attempt to clarify this sentence, but it still does not read clearly – I cannot understand if the authors mean to say that despite being an isolated phenomenon there is over-diagnosis? Or do the authors mean that there is underdiagnosis as the sonographic characteristics of molar ectopic pregnancy are indistinguishable from those of non-molar ectopic pregnancy?

Thank you for your revision, we tried to be clearer: we only would like to explain that a molar pregnancy is an isolated phenomenon in the context of all tubal ectopic pregnancies. We wrote in this way to avoid any confusion: "It is an isolated phenomenon to have a histopathological diagnosis of hydatidiform mole in an ectopic tubal pregnancy, but it could happen23"